# The Potential Release of Chemicals from Crumb Rubber Infill Material—A Literature Review

**DOI:** 10.3390/jox15050159

**Published:** 2025-10-02

**Authors:** Federica Ghelli, Samar El Sherbiny, Giulia Squillacioti, Nicoletta Colombi, Valeria Bellisario, Roberto Bono

**Affiliations:** 1Department of Public Health and Pediatrics, University of Turin, via Santena 5 bis, 10126 Turin, Italy; federica.ghelli@unito.it (F.G.); samar.elsherbiny@unito.it (S.E.S.); valeria.bellisario@unito.it (V.B.); roberto.bono@unito.it (R.B.); 2Federated Library of Medicine “F. Rossi”, University of Turin, via Santena 5 bis, 10126 Turin, Italy; nicoletta.colombi@unito.it

**Keywords:** public health, end-of-life tyres, artificial turfs, synthetic turf, bioaccessibility

## Abstract

End-of-life tyre (ELT) management is still a hot topic due to implications for sustainability and human health. This review aims to summarise the findings concerning the chemicals’ bio-accessibility/availability from the granular tyre-derived infill material used in sport surfaces. We included 14 original research articles and 5 reports (grey literature). The results included the analysis concerning volatile organic compounds (VOCs), polycyclic aromatic hydrocarbons (PAHs), phthalates, metal(loid)s and other substances. The release of some dangerous chemicals was demonstrated, even though results must be considered critically. However, the chemicals’ bioaccessibility shows a highly nuanced picture and is not, *per se*, sufficient to determine the risk for the exposed subjects. The lack of bioavailability and epidemiological studies analysing the exposures in real scenarios resulted in one of the main issues concerning a proper evaluation of the potential risks for human health.

## 1. Introduction

End-of-life tyre (ELT) management, also known as post-consumer tyres (PCTs), is still a hot topic, with important implications for sustainability and human health [1].

In 2021, the 27 EU Member States plus Norway, Switzerland and the UK discarded 4200 kt of ELTs each year [2]. Since these objects are not biodegradable, their abandonment or disposal in landfills represents a significant threat to the environment and public health [3].

Tyres are complex, component-rich products whose composition differs according to their type, application and producer. The main component of vehicle tyres is a mixture of styrene butadiene rubber (SBR), natural and butadiene rubber blended with specific additives to guarantee the tyre performance [4]. When these items are no longer usable for their intended purpose, in most cases they can be employed to recover energy or undergo material recycling. One of the main secondary raw materials obtained thus far are ELT-derived rubber granulates, whose most important application is infill material in synthetic turfs, i.e., surfacing infrastructures engineered to mimic the appearance and the sports performance of natural grass on athletic fields [5]. In 2018, a total of 51,616 synthetic pitches was estimated in Europe, covering an area of 112 million square meters [6], and the installation of 4200 new pitches and 6600 mini-pitches each year was estimated for the decade 2018–2028 [7].

Over the years, technological and material advancements resulted in the improved safety, playability and durability of these surfaces [5]. However, the sustainability of these infrastructures remains controversial. While some authors highlighted potential benefits (e.g., reduced water needs, no pesticides/fertilisers requirements, higher resistance to climate and weather conditions, and recycling of scrap tyres), others pointed out considerable maintenance inputs, costs and ecosystem disservices (e.g., absence of atmospheric C sequestration, O_2_ production and cooling effects, increased flooding risks, soil erosion, stormwater and pollution runoff) [5,8]. Moreover, these areas can easily reach extreme temperatures [9], often requiring a sprinkler irrigation to cool the surfaces [10]. Also, the analysis concerning the risk of injuries are controversial, with some authors highlighting a higher likelihood of heat- and non-contact-related injuries [11,12].

Nowadays, tyre-derived infill material is under the spotlight for both the presence of hazardous substances and the particle size of the rubber granulates [3,4].

Besides rubber, the granulate composition includes several additives, impurities and compounds accumulated or newly generated during vulcanisation or the exposure to environmental conditions [13]. Some of these substances are carcinogens, reprotoxic or able to bioaccumulate [14]. Despite the European Chemicals Agency (ECHA), as well as other agencies operating in the field of health protection and environmental safety are consistent in finding “*no reason to advise the people against playing sports on synthetic turf containing recycled rubber granules as infill material*” [4], the ECHA itself highlighted some knowledge gaps [4], and the revision of the exposure limit considered safe for lifelong exposures is still ongoing [7].

This concern led some international Agencies involved in public health and health protection (e.g., the Food and Drug Administration (FDA), the Centers for Disease Control and Prevention (CDC) and National Institutes of Health (NIH) supporting the National Toxicology Program (NTP); the European Chemicals Agency (ECHA); the National Institute for Public Health and the Environment (RIVM); U.S. Environmental Protection Agency (EPA)) to investigate the presence of toxicants and their potentially detrimental effects [4,13,15,16,17]. The concentration of polycyclic aromatic hydrocarbons (PAHs), volatile and semi-volatile organic compounds (VOCs and SVOCs), benzothiazoles, chlorinated paraffins, polychlorinated biphenyls (PCBs), per- and polyfluoroalkyl substances (PFASs), phthalates, plasticisers, antioxidants, vulcanisation additives and heavy metals can be highly variable, depending on the manufacturing and recycling process, the rubber type, the granulometry, the presence of coatings and the age of the pitch [18,19,20]. Environmental factors and the active use of pitches may enhance the volatilisation, runoff and leachate of these chemicals, resulting in the contamination of sewage, groundwater and natural surface water [18]. These chemicals can be accessible to players and general users through dermal contact, inhalation and ingestion, especially for children [21], due to hand-to-mouth and object-to-mouth behaviours [22]. However, the granule (0.4–2 mm) is big enough to be sensed in the mouth and would be more likely spitted out than ingested [23].

The European Commission recently classified the granular infill material in sports surfaces as “*the largest source of intentionally added microplastics in the environment*” and whose use should be restricted. The ban will enter into force from October 2031 [24] and adds to similar measures (banning, restriction or moratoriums) in force in some US jurisdictions and communities [25]. Micro- and nanoplastics are ubiquitous pollutants in both anthropic and natural environments, making the uptake and accumulation in real-world scenarios a complex chronic process, mainly via inhalation and ingestion [26] and associated with inflammation, oxidative stress, cytotoxicity, genomic instability and immune system dysfunctions [27]. These effects might be related to their physical–chemical properties, concentration, chemical constituents and/or contaminants, or microbial growth [28]. The mechanisms through which toxic chemicals could be adsorbed onto or released from micro-particles are not clearly understood yet, but the ageing of particles and the polymer composition are likely to be involved [28]. Hence, the importance of assessing the release and the bioaccessibility of the different chemicals in ELT-derived infill material and the possible conditions that can affect this phenomenon.

The term bioaccessibility can be defined as the fraction of a chemical that is potentially available for absorption, specifically in the gastrointestinal tract, even though this term is commonly used in the literature in an even broader way [29,30].

The present review aims to describe the state of the art of the current knowledge concerning the potential release in simulated biological fluids of chemicals from the granular infill material used in artificial turf sport fields and playgrounds, considering the results in both the scientific and grey literature.

## 2. Search Strategy and Literature Review

### 2.1. Search Strategy

The search was launched on 16 May 2023 in five databases, i.e., PubMed, Embase, CAB Direct, Scopus and Web of Science, and updated up to 1 September 2025. The quality of the search strings was tested with gold-standard articles. To maximise accuracy, we included a wide list of keywords concerning the sport surfaces and the granular infill material (e.g., “artificial turf”, “synthetic turf”, “crumb rubber”, “infill material”, etc.) and the potential adverse effects for health (e.g., “health”, “toxicity”, “bio-accessibility”, “biofluids”, etc.). According to the snowball search method, the bibliography of the included articles was screened to find relevant articles not retrieved by the search string. The search was also extended to the grey literature, whose records were identified by a proper search on Google and Google Scholar and was implemented through the snowball methods as of the reference lists of the included research articles.

As reported in the Third International Conference on Grey Literature in 1997, the term grey literature consists in the information “*that which is produced on all levels of government, academia, business and industry in electronic and print formats not controlled by commercial publishing*”.

The full search strings are provided in Appendix A, as well as the list of abbreviations used.

### 2.2. Eligibility Criteria

The eligibility criteria to select the records to be included in the present review were (i) original research focusing on release, bioaccessibility/availability of toxic components from the rubber granulate able to affect human health; (ii) grey literature concerning release, bioaccessibility/availability reporting original measurements, or elaborations provided by agencies involved in the protection of human health and the environment. Only full texts in English were considered suitable. Full text with unpublished data, reviews, expert opinions, editorials, protocols and conference abstracts were excluded. After duplicate removal, two independent reviewers (F.G. and S.E.S.) performed the article screening selection while blind, at first based on titles and abstracts and then on the full texts of the selected records. Discrepancies were discussed with a third reviewer (G.S.). When scientific articles and reports were reporting the same data, we prioritised the scientific literature (excluding the reports) to avoid duplicate records.

### 2.3. Data Extraction

Two independent reviewers (F.G. and S.E.S.) extracted the following data from the selected documents, and reported them in a spreadsheet: country, aim, study design, analytical methods, exposure, matrix, analysed markers and main results.

### 2.4. Visualisation and Data

Data visualisation has been realised using BioRender.com. Graphical abstract has been created in BioRender. Squillacioti, G. (2025) https://BioRender.com/2ci60m6 (accessed on 22 September 2025).

In Appendix A, the minimum and maximum concentrations/rate percentages shown in scientific articles are reported.

## 3. Results

### 3.1. Qualitative Synthesis

Among the 8650 research documents initially identified, 3769 duplicates were removed, and the remaining 4881 items were subjected to the multi-step screening process. In the end, 14 original research articles were included in this review [14,31,32,33,34,35,36,37,38,39,40,41,42,43]. Concerning the grey literature, the research on Google/Google Scholar provided 59 items, while an additional 55 were identified through the snowball search method. At the end of the screening process, we included 5 items [16,17,44,45,46]. The multi-step process is synthesised in the PRISMA diagram reported in Figure 1.

Figure 2 shows the records’ uneven distribution worldwide. A summary of the studies included is reported in Appendix A, while in Appendix A there is a summary of the bioaccessibility measurements from the scientific literature.

Among the included items, three research articles and two reports referred to data concerning playgrounds or school playgrounds [33,34,42,44,46], while the others refer to crumb rubber from a synthetic turf playground or local provider and destined for artificial turf playgrounds. A visualisation of the percentage of studies analysing the different groups of chemicals in the scientific and grey literature is reported in Figure 3.

### 3.2. Bioaccessibility Study Characteristics as Reported by Scientific Literature

Research studies were performed from 2008 to 2025 in USA (*n* = 3), Korea (*n* = 2), Japan (*n* = 3), China (*n* = 2) and Europe (*n* = 4). The matrix employed were crumb rubber or rubber chips from recycled tyres. Only a few studies specified the rubber type, like ethylene propylene diene monomer (EPDM—[33]) or mixtures of different polymers, such as thermoplastic elastomer (TPE), EPDM, styrene butadiene rubber (SBR) and natural rubber (NR) [35,37].

The most investigated markers were metals (*n* = 9), followed by PAHs (*n* = 5) and other compounds (*n* = 4) (Figure 3). Regarding the treatment employed to prepare and extract the analytes from the matrix before the bioavailability tests, the most used was the digestion in biofluids (*n* = 11) using different techniques. Four studies used solid-phase extraction (SPE) or direct solid-phase microextraction (DI-SPME) to isolate the compounds obtained from the in vitro digestion before the analysis by gas chromatography coupled with mass spectrometry (GC-MS). This technique was used for the analysis of PAHs, VOCs and phthalates. Two studies employed the BARGE-UBM method to assess the specific bioaccessibility in case of accidental ingestion. The most used biofluids were gastric fluids (*n* = 13), followed by saliva (*n* = 9), intestinal or duodenal juices (*n* = 11) and sweat (*n* = 7). Other biofluids investigated were bile (*n* = 2) and lung solution (*n* = 1). The quantifications were also performed in the evaporated air near the headspace (*n* = 1) and in filters simulating children touching (*n* = 1).

Despite the analytical methods employed being strictly dependent on the markers analysed, they were quite homogeneous among studies. GC–MS and LC–MS/MS (*n* = 7) were employed to quantify PAHs, phthalates, (S)VOCs, and other substances, while inductively coupled plasma mass spectrometry (ICP-MS) analysis (*n* = 9) was employed to quantify metals. One study utilised the high-performance liquid chromatography with diode-array detection (HPLC–DAD) technique to quantify benzothiazoles and amines. In one study, the HPLC technique coupled with a fluorescence detector was employed for the quantification of PAHs.

### 3.3. Volatile Organic Compounds (VOCs) as Reported by Scientific Literature

Pronk et al. [14] measured the release of VOCs in the evaporated air from rubber granulate under warm weather conditions. The VOC concentrations measured were below the LOD or very low. The results seemed to demonstrate that the inhalation route is not a relevant exposure route due to the low concentration of these chemicals.

### 3.4. Polycyclic Aromatic Hydrocarbons (PAHs) as Reported by Scientific Literature

Armada et al. [31] demonstrated the bioaccessibility in gastrointestinal fluids of 15 out of the 16 PAHs classified as priority pollutants from US-EPA (except for D[ah]A), plus benzo[j]fluoranthene and benzo[e]pyrene, listed as carcinogenic by the ECHA. Analyses were performed in crumb rubber samples from both infill material collected from football pitches across Europe (Poland, Portugal and Sweden), and purchased from local suppliers (Portugal and Spain). In both the rubber granulate and the bioaccessible fractions, the compounds with higher concentrations were the heaviest and least volatile, with similar profiles between the two fractions. The volatile PAHs showed the highest bioaccessibility rate. Moreover, phenanthrene (PHN), classified as a substance of very high concern (SVHC) and dangerous when ingested, was bioaccessible in all samples up to 60 ng/g (bioaccessibility 0.25%), while B[α]P, which is carcinogenic, mutagenic and reprotoxic, was detected in 75% of the bioaccessible fraction up to 2.5 ng/g. The daily uptake of all targeted PAHs by children (3–6 years) was estimated from 0.16% to 0.90% of the oral reference dose proposed by US-EPA for B[α]P (300 ng kg^−1^ day^−1^). The author hypothesised that the lower particle size and the young age of pitches could result in a higher bioaccessibility. Other studies revealed only some positive results in their analysis. Pronk et al. [14] analysed 546 infill material samples from outdoor turf pitches in the Netherlands. According to their estimates, at 60 °C, the maximum B[α]P concentration in the air above the sport surface could be estimated at 0.03 ng/m^3^. Migration tests revealed that only a fraction of the PAHs in the infill material can migrate in biofluids; specifically, 9% can be found in artificial gastrointestinal juice, and only 0.02% in sweat. The author highlighted a robust association between both content and migration level. Schneider et al. [39], analysing the migration of PAHs from ten samples of infill material in sweat, revealed that the concentration of these chemicals could be found only in some samples, and the most common were fluoranthene, pyrene and naphthalene. Except for chrysene in one sample, none of the eight REACH PAHs was detected. Zhang et al. [43] analysed the bioaccessible fraction in a selection of samples from different sport fields in New York, differing in brand name and age of the turf. In digestive fluids, the authors highlighted a wide variety in naphthalene, benzo(α)pyrene and benzo(ghi)perylene bioaccessible fractions. These last two compounds were detected only in gastric fluids in only one of the investigated samples, with a bioaccessibility lower than 3%. Naphthalene was instead also detected in saliva and intestinal fluids. Moreover, the PAH content was found to decline over time. On the contrary, Nishi et al. [37], analysing elution samples from industrial rubber and discarded tyres, did not find any of the analysed PAHs at concentrations above the limit of quantification (LOQ). Pavilonis et al. [38] also found, analysing the extraction of chemicals from new crumb rubber infill samples and actual field samples in sweat, lung and digestive artificial biofluids, that the 16 targeted PAHs were mostly under the LOD in biofluids. Maximum concentrations were higher than the LODs only for some compounds in both lung and digestive fluids (benzo[α]pyrene, benzo[b]fluoranthene, benzo[k]fluoranthene, chrysene, dibenzo[a,h]anthracene). The concentration extracted varied according to the artificial matrix employed, with the highest values in sweat and the lowest in digestive fluids.

### 3.5. Phthalates as Reported by Scientific Literature

The migration test performed by Pronk et al. [14] revealed that 20% of phthalates are able to migrate in gastrointestinal solutions, with a robust relationship between the chemical content and the migration level. Schneider [39] found phthalate concentrations above the LOQ in almost half of the sweat samples.

### 3.6. Other Components as Reported by Scientific Literature

Some studies revealed the presence of some other volatile compounds. Armada et al. [31] detected 6PPD, HMMM and 6PPD-quinone in all biological fluids, plus BTZ and MBTZ in some of them. Kawakami et al. [32] assessed the elution of 38 compounds (vulcanisation accelerators, antioxidants, decomposition products of curing agents, vulcanisation retarders, peptising agents, plasticisers, light stabilisers, and others) from eight crumb rubber samples. In total, 27 of them were measurable in one or more than one samples. MBT, ETU and AP showed an elution rate ≤ 10%, in contrast with BTZ, BZL, TEP and PI showing up to 90%. Plasticisers revealed an elution rate near or <LOQ. Higher detection frequencies were measured in simulated gastric juices, especially for antioxidants, probably due to their basic properties. Pavilonis et al. [38] identified 4-tert-ocyl phenol in lung biofluid (0.2 mg/kg) and in sweat (1.0 mg/kg), and 2,2-benzothiazole in digestive fluid (10 mg/kg). Schneider et al. [39] highlighted the presence of amines (tert-butylamine) and benzothiazoles, even though the high LOQs and possible interference with the rubber may have limited the possibility of measuring these substances. Specifically, some amines (DPG and 6PPD) and benzothiazoles (MBT and 2-hydroxybenzothiazole) seemed difficult to be detected. MIBK and 4-tert-octylphenol were found in all the artificial matrices investigated, i.e., sweat, saliva and gastric fluids. In this study, amine, benzothiazole compounds, cyclohexanone and MIBK were characterised by high migration rates. Migration in artificial sweat was also measured for BPA and DINP, even though they are present in rubber granulates only in small concentrations. Soñora et al. [40] assessed the bioaccessibility in saliva, bile, gastric and duodenal juices of eleven chemicals present in tyres, including antiozonants, vulcanisers and a crosslinking agent in samples from seven football fields, a kids’ playground and two commercial providers. All the compounds in rubber samples were detected in the corresponding bioaccessible fraction, except for compounds at extremely low concentrations in crumb rubber. DMBA, 6PPDq and BTZ were measurable in all the biofluids, this last being the most abundant in all the samples and in concentration up to 330 μgL^−1^ and showing the highest % bioaccessibility, mostly ranging between 20% and 50%, with a peak of 73%. DMBA showed a mean bioaccessibility value of 20%, followed by DPG (16%), MBTZ (13%), DMBA (10%), CBS (5.6%), DTG (5%), IPPD (4.9%), 6PPDq (1.8%), DPPD (0.6%) and 6PPD (0.2%).

### 3.7. Metal(Loid)s as Reported by Scientific Literature

Pavilonis et al. [38] found that the vast majority of target metals were below the LOD, with a higher percentage of censored observations in the extracts from field samples (71%) than from the new infill samples (60%). As, Be, Cd, Se and Ag were under the LOD in all or in the majority of the samples for all the biofluids. Pb, Ti and V were the most common, and Pb was detected in almost all field samples in both digestive fluids (up to 260 mg/kg) and sweat. Kubota et al. [35] performed the extraction test on the six infill samples with the highest metal concentration among those retrieved from ten synthetic turf companies. The detection on simulated biofluids was different according to the sample and the biofluids considered. The artificial gastric fluid was found to be the artificial biofluids where was possible to retrieve the majority of highly-concentrated and dangerous metals. Al, Fe, Mn and Zn were the most common, with Zn being highly concentrated in all matrices. The extraction rate was higher than 10% only in gastric fluids. In this matrix, Pb was detected quite frequently (38%) with an extraction rate of 4.3%, while Ba, Cr and Sb were detected with an extraction rate of, respectively, 6.9%, 5.6% and 0.08%. Also, Pronk et al. [14] revealed that metals can be retrieved in gastrointestinal juices and sweat, while highlighting the poor likelihood of migration of these chemicals into water. The most leaching metals were found to be Zn, Cu, Co and Ba; even these last with a migration 150–560-fold lower than Zn. The authors assumed that skin exposure to metals in playing on artificial sports surfaces during or after rainfalls could be considered negligible. Kim et al. [34] revealed that the metal extraction with acid and artificial gastric solutions from infill chips resulted in concentrations ranging between 10 and 10,000-fold lower than the total content in rubber. Specifically, the bioaccessibility in gastric solutions was estimated to be up to 41.2% for Pb and 8.99% for Cr. The bioaccessible fraction for Cd and Zn were estimated to be extremely low, 100 and 10,000 times lower than the total content level, respectively. A comparison between the acid extraction and the extraction with gastric solution revealed approximately a 10 times higher concentration in the biofluids of “softer” metals such as Pb and Cr, while similar levels for “harder” metals, such as Cd and Zn. Zhang et al. [43] highlighted a wide variety between samples in the bioaccessible fractions in gastric fluids concerning metals, specifically Pb and Cr. Cr was also detected in saliva. The Pb bioaccessibility ranged between 24.7% and 44.3%. Schneider et al. [39] highlighted that Co and Al can be found in sweat, saliva and gastric juice, even though not in all samples. Kim et al. [33], aiming to estimate the potential exposure to Pb in artificial sport surfaces through ingestion, revealed a significant difference according to the particle size, with higher exposure in younger subjects exposed to particles smaller than 250 µm. Tian et al. [41] analysed 162 dust samples from different kinds of outdoor sport facilities in 17 campuses in Beijing. Data for bioavailability were presented as aggregated and are not referred only to synthetic turf. Leaching experiments in simulated sweat and gastric juice revealed that in this last matrix the metal(loid)s is more effective in enhancing the process, with concentrations ~5–25 (interquartile range) times higher than in sweat. The prolonged exposure to dust is related to a higher metal(loid) bioavailability. Moreover, the concentrations of these chemicals in simulated biofluids were 1–2 times higher than in simulated rainwater. As (III) was retrieved from simulated sweat exposed to dust, probably due to blue/green pigments containing Cu-As. Luo et al. [36] measured the heavy metal bioaccessibility in a simulated gastric and gastrointestinal environment from recycled tire crumb with different ages from northern and southern regions of China. The concentration of Zn, As and Cd were found to decrease from the oldest (ten years) to the newest (one year) and were higher in samples from the northern regions. In the gastric phase, the As bioaccessibility coefficient from northern samples with a construction time of ten years was higher than 0.20 (lower limit for bioavailability in this study), while this value was lower than the threshold for Zn, As and Cd from samples with a lower construction time. The bioaccessibility coefficient values in the intestinal phase were higher than in the gastric phase, with As ranging between 0.20 and 0.40. Winz et al. [42] estimated the amount of metals available through dermal contact on a playground made from tire crumb rubber tiles through surface wiping, highlighted that the measured concentration was positively correlated with the concentration in the crumb rubber. Moreover, Pb was found to have a bulk content ~3 times higher than Ba and Cr, but the surface release measured was ~4 times lower. Colour additives were not found to be determining.

### 3.8. Bioaccessibility Study Characteristics as Reported by Gray Literature

The publication dates range from 2009 to 2019. Four reports are from the USA, namely redacted by Environmental Protection Agency (EPA) (*n* = 2), Environmental and Occupational Health Sciences Institute Robert Wood Johnson Medical School (*n* = 1) and the California Environmental Protection Agency’s Office of Environmental Health Hazard Assessment (OEHHA) (*n* = 1), while one report is from the Netherlands (RIVM). All reports that have been included used crumb rubber as matrix for their bioavailability studies.

### 3.9. Volatile Organic Compounds (VOCs) as Reported by Gray Literature

According to Groot’s [16] report, benzene, toluene, ethylbenzene, xylene, styrene and 1,3-butadiene were not detected in the evaporated air, even at high temperature (60 °C). Ethanol, acetone, acetaldehyde, carbon disulphide, methyl ethyl ketone and methyl isobutyl ketone were found in limited concentrations. Exposure through inhalation was considered negligible, due to the limited release of these substances into the air.

### 3.10. Semi-Volatile Organic Compounds (SVOCs) as Reported by Gray Literature

Researchers of the OEHHA [46] were able to quantify cyclohexanamine N-cyclohexyl, cyclohexanone, formamide N-cyclohexyl, 1H-isoindole-1,3 (2H)-dione, o-cyanobenzoic acid, aniline, 2(3H)-benzothiazolone, and phenol in the artificial gastric juices. According to Groot [16], exposure to PAHs was mainly due to ingestion: 9% of these chemicals was found to be able to migrate in the artificial gastrointestinal fluids, while varies between 0.001 and 0.002 in sweat. Lioy [45] reported that all the investigated chemicals in artificial biofluids were below the LODs, except for Acenaphthylene (0.08 mg/kg in one lung extraction).

### 3.11. Metal(Loid)s as Reported by Gray Literature

According to OEHHA [46], simulated gastric digestion resulted in the quantification Sb, Mb, V, Ba, Cu and Zn. The analysis of Highsmith et al. [44] revealed that in the digestive fluids the Pb bioaccessibility ranged from 1.6% to 10.7% from synthetic turf infill, from 0.2% to 86.8% from blades, and from 0.3% to 7.4% from playgrounds. For blades and playgrounds, the lowest bioaccessibility values corresponded to the highest total extractable Pb. The Pb bioaccessibility seemed to be influenced by the blades colour in the following orders black > white and green > red. Lioy et al. [45], reported that Be, As, Se, Ag and Cd were found to be below the LODs in all or in the majority of the samples. Pb, Cu and Mg were found to be most concentrated in digestive fluids, while V and Cr were found to be mostly concentrated in artificial sweat, followed by lung and then digestive fluids. The analyses of Groot et al. [16], following exposure to rubber granulate, highlighted that Pb and Co can be detected in gastrointestinal juices and sweat (Pb 0.07 µg/g and 9 µg/g, and Cb, 0.48 µg/g and 2 µg/g, respectively), while Cd only in artificial sweat (0.02 µg/g). In its recent report EPA [17] confirmed the highest bioaccessibility in gastric biofluids of metals from infill samples, followed by sweat with sebum, and saliva. In these three biofluids, the metal with the highest median concentration was Zn (129 mg/kg, 11 mg/kg and 0.72 mg/kg, respectively). Considering the different matrices, Zn, Fe and Mg were the most abundant metals in gastric juices Zn, Mg and Cu in sweat plus sebum, while Zn, Mg and Co in saliva. Mn was the metal with the highest median percentage in gastric juices and in sweat with sebum (12% and 1.5%, respectively), while Mg was the one with the highest median bioaccessibility in saliva (0.2%). Pb resulted to have the highest bioaccessibility in gastric biofluids (0.29 mg/kg, 1.9%), followed by sweat with sebum, and saliva (0 mg/kg, 0% in both cases), with a detection rate of 100%, 12% and 22%, respectively. Moreover, the percentage of bioaccessible Pb was found to be higher in synthetic turf infill samples than in crumb rubber samples from the recycling plants (0.54 vs. 0.18 mg/kg, 3.2% vs. 1.8%). Splitting the data according to the granulate source, the simulated digestion of samples from artificial fields revealed higher concentration of Al, Co, Pb and Ni, as well as in saliva concerning Mg and Sr. In analysing the metals’ bioaccessibility, the heterogeneity among samples was found to be noteworthy. In the report of Highsmith et al. [44], the authors highlighted up to a 4-fold difference between aliquots from the same sample, and up to 36-fold among crumb particles of the same playground. Also, according to Lioy et al. [45], Pb was found to have more than a 2-fold variability in the content in particles, and an even greater variability concerning the load of the surface particles.

### 3.12. Phthalates as Reported by Gray Literature

According to the report of Groot et al. [16], the phthalates concentration in artificial sweat were below LOD. Instead, the 20% of phthalates present in rubber granulate was retrieved into artificial gastrointestinal fluids.

### 3.13. Other Components as Reported by Gray Literature

Lioy [45] reported the presence of 4-(tert-octyl)phenol and butylated hydroxytoluene (BHT) in artificial lung biofluids and sweat samples.

## 4. Discussion

The potential risk for health due to the exposure to tyre-derived infill material in sport surfaces remains a matter of concern. Indeed, despite local and international measures restricting the use of this material (e.g., the restriction on force in Europe to reduce the microplastic pollution in the environment [47]), the existing sport facilities will continue to be employed. Moreover, these measures will not be implemented worldwide [20].

Artificial turfs are widely used for both recreational and professional purposes and, as accessible sport settings, they can play a critical role in promoting the engagement in regular physical activity and exercise [48]. This is relevant since more than 30% of the world’s adult population and more than 80% of the adolescents (aged 11–17 years) are estimated to be physically inactive, i.e., not meeting the WHO’s recommendations, with not only enormous consequences for their and their’ families health, but also a large economic burden for health services and society [49,50,51,52].

However, the implementation and the use of these infrastructures remain highly controversial, even in the scientific field. The concern is whether artificial turf fields and playgrounds could be considered safe environments, especially for children that may be at higher risk due to a higher vulnerability to heat, higher metabolism rates and a higher susceptibility of their developing target organs [53]. To date, this is a hot topic with many research groups trying to highlight the various possible detrimental health effects related to the exposure to these surfaces, resulting in a general call for caution [54]. However, the approaches employed in the studies implemented thus far are highly heterogeneous, and there is still a stark contrast between the large body of literature assessing the presence of noxious chemicals in the components of artificial turfs and the paucity of studies systematically investigating the potential health effects associated with real-life exposure scenarios [55]. A recent systematic review, updated to 2023, addressed this topic, but in the last two years, new articles were published reporting some new interesting results. Moreover, this publication is only related to the scientific literature and does not include the results of the grey literature [56].

Also, the current literature analysing the potential in vivo effects in humans are still rare and heterogeneous in terms of both chemicals and matrix under investigation. Van Rooij et al. [57] reported that the uptake of PAHs was minimal by assessing the concentration of 1-hydroxypyrene in the urine of seven football players training and having a match on artificial turf. Huang and colleagues, analysing the presence of environmentally persistent free radical (EPFR)-carrying crumb rubber in saliva samples of 200 participants, revealed a potential health risk for the individuals exposed, since this substance can be related to the inhibition of salivary amylases and salivary lysozyme activity [58]. Finally, [59], comparing the metal concentration (Hg, As, Al, Pb, Zn, Mg, Fe) in blood samples from 17 field hockey athletes having a tournament on synthetic turf playgrounds with 11 controls, reported that even though some differences were observed, these were still within the international reference ranges. However, in this last case, the authors did not measure the corresponding metal concentration in the infill from the artificial playground [59]. In other cases, the measured biological effects were related to the assessment of differences in the sport performances [60]. A recent report of EPA reported the results of a biomonitoring study, measuring metals and PAH biomarkers in the blood and urine of youth and adult soccer or American football players [61]. This publication revealed blood Se levels higher than the geometric mean of the general population, even though the concentration of this substance in crumb rubber was lower than the LOD. The levels of the other metals were in line with the results for the general population. Concerning the urinary PAHs, the authors observed a significant difference between pre and post-activity only for 2-hydroxynaphtalene. The adjusted creatinine levels were found to be similar to the levels observed in the general population for most of the compounds under investigation.

Concerning the analysis performed on crumb rubber, despite that factors such as particle size, age of the pitch, climatic conditions, and maintenance operations are supposed to play a role in enhancing the release of these chemicals [31,62,63,64], only a few studies considered these variables. In addition to the differences in methodological approaches, these issues represent the main limitations and biases of the included studies, eventually affecting the data reliability. The authors often did not provide an estimation of the possible biases affecting their research, limiting their discussion to a general overview of the possible limitations and preventing the estimation of the role of the infill characteristics in determining its properties. Concerning the characterisation of the chemical profile, there is significant variability in the compounds under investigation, resulting in difficulties in the results comparison [40].

Moreover, the studies reporting exposure measurements are sparse, and no epidemiological studies have been carried out yet [21].

Despite an apparent homogeneity in the particle size in real pitches [31], the possible presence of small-size particles was reported [33]. The increased ratio between surface area and mass in the smaller particles affects their toxicity and their ability to interact with biological systems [26]. Moreover, the interaction with the environment may result in an enrichment of the granule chemical composition. The decreasing particle size was found to be related to a higher HQ [33] and a higher PAH bioaccessibility [31], while it did not seem to be associated with the metal content [63].

Outdoor synthetic turfs face several environmental conditions and stressors, gradually degrading the granulate polymeric structure and affecting the chemical release [35,65]. Concerning temperature, an ambient air of 25 °C with direct exposure to sunlight corresponds to a surface temperature of 60 °C, with possible peaks above 90 °C, enhancing the release of water-soluble compounds that can then be spread in the environment [66]. The PAHs in the runoff may represent a risk for dermal contact and an ecological risk for water bodies and the surrounding environment [67]. The degradation due to environmental factors was also associated with a higher metal leachability [35], and rainfall events may also affect the chemical profile of the leachate [67]. The pitches used may also easily accelerate the ageing process. However, the high variability in the chemical profile among different turfs is primarily related to the various crumb rubber sources, and can be observed even among samples from the same site [38]. This difference can be exacerbated over time due to the periodic refill of the granulate matter, creating a unique blend for each field and not reflecting the field age [43,63]. Consequently, real-world scenario measurements may only be generalised to pitches of the same area and age [38,63].

One of the most evident results of the present review is the high heterogeneity in findings from both the scientific and grey literature. The authors employed different methodological approaches and theoretical assumptions, preventing the generalisability of results and the comparison between both results from different studies and the thresholds recommended by the agencies involved in health and environmental protection. Moreover, the vast majority of the studies analysed the characteristics of a specific site [63]. Significant heterogeneity can also be observed in the chemicals under investigation. The lack of data and the controversies among the results can be due to the attempt to summarise the patchy literature.

The characterisation of the chemical profile of artificial biofluids following the exposure to crumb rubber granulate is of the utmost importance to assess the real concentration of hazardous substances interacting with biological systems [31]. However, these data are still limited, and it is important to be cautious in their interpretation [31]. Further studies are necessary to analyse possible synergic effects among different compounds in complex mixtures and the eventual outcomes on the human body [68]; e.g., there could be PAHs other than the REACH 8 PAHs with genotoxic properties, leading to a risk underestimation [7].

Moreover, the bioaccessibility varies among the artificial biofluids according to their properties. The low pH in gastric fluids may be responsible for a higher metal concentration and a lower organic compound concentration compared to other matrices [29,34]. The poor metal solubility in lung fluids instead resulted in negligible exposures to these chemicals through inhalation [38]. However, the measurements of the total amount released in migration studies might lead to an overestimation of the amount of chemicals realistically available for the human body, since lipids and suspended matter may reduce their absorption [14].

In artificial turfs, the infill material is not the only source of hazardous chemicals. For example, the dyes employed to paint the synthetic grass fibres may be a source of Pb and Cr [38], making difficult to identify the proportion of hazardous substances originating from the tyre mixture [63].

Conversely, the use of data concerning migration in water-based sweat might underestimate the real exposure to lipophilic substances such as PAHs [14]. Also, the traditional PAH bioaccessibility assessment in the digestive tract does not consider the lipidic component and the possible interaction with food, possibly enhancing PAH bioavailability [43]. To overcome this issue, Schneider and colleagues employed an aqueous ethanol-based sweat simulant [39].

However, it is worth noting that the bioaccessibility results, even if showing a high nuanced picture, are not per se sufficient to determine the risk for the subjects exposed, and this synthesis should be considered cautiously due to methodological differences and biases that could have affected the results.

The main strengths of this review lie in the employment of a wide search string to increase the search sensibility, and the inclusion of a systematic search in the grey literature to extend the plethora of possible results, allowing us to point out issues and criticisms affecting the approach in assessing the potential threat due to the exposure to rubber granulate in artificial turfs. The main limitations are instead represented by the scattered approach of both the scientific and grey literature in addressing the issue and preventing the possibility of reaching unambiguous conclusions.

## 5. Conclusions

This review highlighted the potential release of some dangerous chemicals from crumb rubber, even though results must be considered critically, since different experimental protocols have been used and several biases could have affected the results. Moreover, the potential risks for health were in some cases reported only in worst-case scenarios, and these results are not per se sufficient to determine the risk for the subjects exposed. The literature analysis revealed a stark contrast between the body of literature assessing the presence of noxious chemicals in crumb rubber and a small number of studies systematically investigating the potential health effects associated with real-life exposure scenarios, preventing a proper evaluation of the potential risks for human health.

A holistic approach to monitoring the eventual modulation of biological outcomes as a result of the variation in the composition of artificial turf sport venues and playgrounds in real exposure scenarios would be highly recommended.

## Figures and Tables

**Figure 1 jox-15-00159-f001:**
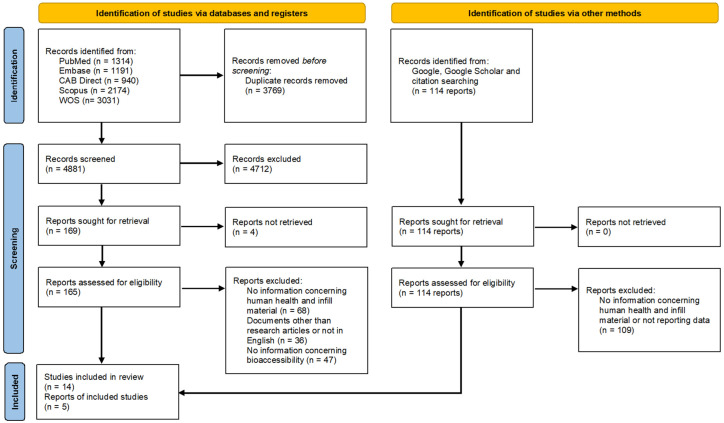
Summary of the search strategy.

**Figure 2 jox-15-00159-f002:**
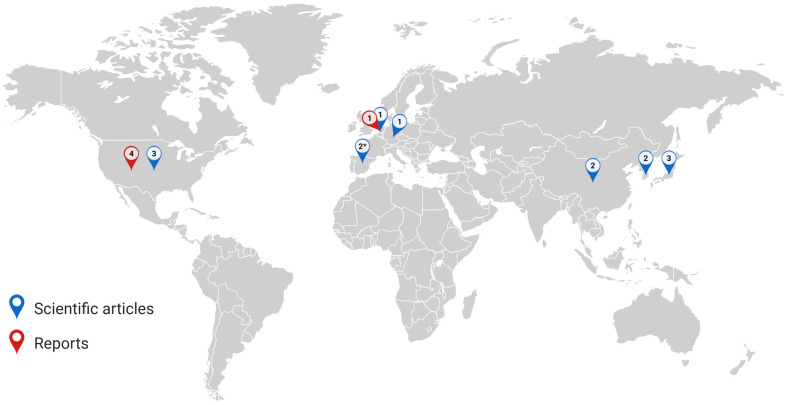
Depiction of the records’ uneven distribution. As can be observed, the data are not representative of the worldwide situation and are limited to a reduced number of countries. Numbers represent the number of studies/reports included in the review. The asterisk (*) refers to the study of Armada et al. [30] that is based on data from artificial turfs in Spain, Poland, Portugal and Sweden. Created in BioRender. Squillacioti, G. (2025) https://BioRender.com/nj4hahb (accessed on 22 September 2025).

**Figure 3 jox-15-00159-f003:**
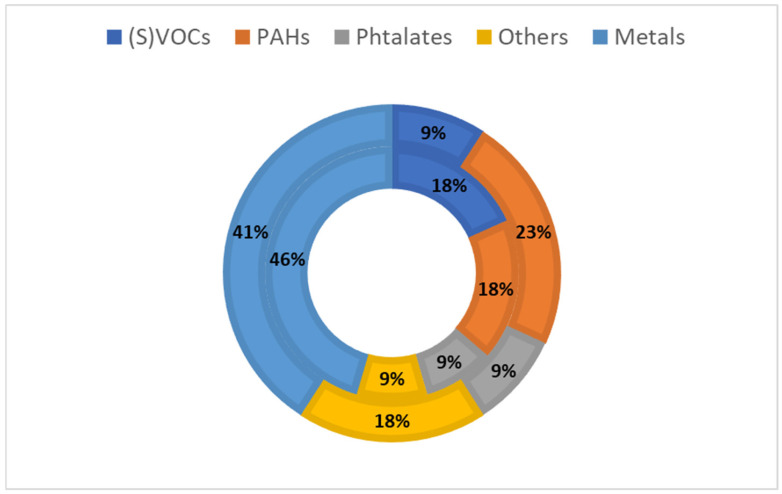
Visualisation of the percentage of studies analysing the different groups of chemicals in scientific literature (outer circle) and grey literature (inner circle).

## Data Availability

No new data were created or analyzed in this study.

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
