# Peer review of "The Potential Release of Chemicals from Crumb Rubber Infill Material—A Literature Review"

_jox, 2025, doi:10.3390/jox15050159_

Round 1

Reviewer 1 Report

Comments and Suggestions for Authors

The authors did a great job summarizing what is known about crumb rubber up to 2023. Unfortunately, the research on this topic is advancing rapidly and there are new references and studies that are not included. Particularly (but not limited to), two studies, US EPA and California OEHHA, released in 2024 and early 2025 respectively. 

The authors recommended studies in real exposure scenarios which is a science gap at the moment. However, in their search they did not include the only reference which assessed exposure to PAHs (by using hydroxypyrene in urine as a marker) in players under real playing conditions. See reference below.

Joost G M van Rooij  1 , Frans J Jongeneelen. Hydroxypyrene in urine of football players after playing on artificial sports field with tire crumb infill. Int Arch Occup Environ Health 2010 Jan;83(1):105-10.

I recommend that authors update their literature search up to July, 2025. Pertinent references should be included and analyzed for this paper to be a contribution to the body of research on this topic.

Author Response

Comments and Suggestions for Authors – Reviewer 1

The authors did a great job summarizing what is known about crumb rubber up to 2023. Unfortunately, the research on this topic is advancing rapidly and there are new references and studies that are not included. Particularly (but not limited to), two studies, US EPA and California OEHHA, released in 2024 and early 2025 respectively. 

Thank you for the positive comment. We completely agree that the review needed to be updated.

We thus re-launched the search strings, and we updated the manuscript with the new articles that met the eligibility criteria. The number of articles included is now 14, and the number of reports is 5.

Concerning the US EPA report, it does not contain bioaccessibility data, while for the California OEHHA thus far the report available are still in the form of draft and panel meeting. We preferred to not include them since some possible amendments could still be possible. 

The authors recommended studies in real exposure scenarios which is a science gap at the moment. However, in their search they did not include the only reference which assessed exposure to PAHs (by using hydroxypyrene in urine as a marker) in players under real playing conditions. See reference below.

Joost G M van Rooij  1 , Frans J Jongeneelen. Hydroxypyrene in urine of football players after playing on artificial sports field with tire crumb infill. Int Arch Occup Environ Health 2010 Jan;83(1):105-10.

Thank you for raising this issue, allowing us to better clarify this point. The article was not included since we specifically focused our attention on the bioaccessibility of the chemicals present in the crumb rubber. The mentioned study measured the concentration of PAHs in urine samples from football players active on artificial grounds with crumb rubber infill material reporting that the PAHs uptake is minimal. This conclusion was drawn based on four samples, among which one reported an abnormal PAHs level due to a relevant amount of this chemical assumed through the diet. As reassuring as it may be, these results seem to be unrepresentative of the phenomenon under investigation.

However, in updating our review, we found other similar examples and, for the sake of completeness, we implemented the discussion paragraph as follows: 

“As well, the current literature analyzing the potential effects in vivo in humans are still rare and heterogeneous in terms of both chemicals and matrix under investigation. Van Rooij et al. [57] reported that the uptake of PAHs assessing the concentration of 1-hydroxypyrene in urine of seven football players training and having a match on artificial turf was minimal. Huang and colleagues, analyzing the presence of environ-mentally persistent free radicals (EPFRs)-carrying crumb rubber in saliva samples of 200 participants, revealed a potential health risk for the individuals exposed since this substance can be related to the inhibition of salivary amylases and salivary lysozyme activity [58]. Finally, Gullu et al., comparing the metals concentration (Hg, As, Al, Pb, Zn, Mg, Fe) in blood samples from 17 field hockey athletes having a tournament on synthetic turf playgrounds with 11 controls, reported that even though there some differences were observed, these were still within the international reference ranges. However, in this last case the authors did not measured the correspondent metal con-centration in the infill from the artificial playground [59]. In other cases, the biological effects measured were related to the assessment of differences in the sport performances [60]. A recent report of EPA reported the results of a biomonitoring study, measuring metals and PAHs biomarkers in blood and urine of youth and adult soccer or American football players [61]. This publication revealed blood Se levels higher than the geometric mean of the general population, even though the concentration of this substance in crumb rubber was lower than the LOD. The levels of the other metals were in line with the results for general population. Concerning the urinary PAHs, the authors observed a significant difference between pre and post-activity only for 2-hydroxynaphtalene. The creatinine adjusted levels were found to be similar to the levels observed in the general population for most of the compounds under investigation.”

I recommend that authors update their literature search up to July, 2025. Pertinent references should be included and analyzed for this paper to be a contribution to the body of research on this topic.

We really appreciated the suggestions and comments. We did our best to implement the manuscript up to September 2025.   

Reviewer 2 Report

Comments and Suggestions for Authors

The research strategy is well-structured, although only a few studies were identified. However, this manuscript has some weaknesses that diminish its value as a scientific review. Let's see then:

1) The results should emphasise the most relevant information (the supplementary material shows the results in detail). Then, the main findings should be focused, providing a concise overview for the reader, perhaps by figures. Are there differences in the results obtained in scientific literature versus grey literature? Were all the bioaccessibility analyses performed on simulated biofluids? And about the grey literature (route of exposure – Supplementary table)? The different matrices influenced the bioavailability studies' results (e.g. metals, PAHs)?

2) The discussion section should contextualise and discuss the observed results, relating them to existing knowledge. The authors showed an active role in their criticism and future research.

3) The conclusion should briefly summarise the manuscript, emphasising its significance and suggesting future research directions (in this section, the authors recommended real exposure scenarios).

4) Line 9-10: I think that sustainability integrates environmental preservation...

5) Line 65-66: This concern led some international Agencies involved in public health and health. Exemplify with the name of some agencies

5) Line 102: The quality of the search strings was tested using gold standard articles. What criteria did the authors of "gold standard articles" follow? Which reliable method has been used? An exhaustive list of keywords concerning the sport surfaces and the granular infill material...

6) Line 120-124: Are the authors the independent reviewers? Who are the three independent reviewers?

7) Supplementary table S.2: I propose including in the reference column not only the author's name but also the reference number.

10) line 154: please write: bioaccessibility measurements from scientific literature

11) lines 164, 165: Among the included items, two research articles and two reports referred to data concerning playgrounds or school playgrounds [32,33,37,39], which means that the remaining articles refer to a synthetic turf rubber granule infill (e.g. KUBOTA) or turf sport fields as described in the objectives (?)

Author Response

Comments and Suggestions for Authors - Reviewer 2

The research strategy is well-structured, although only a few studies were identified. However, this manuscript has some weaknesses that diminish its value as a scientific review. 

Thank you for the comment. We were quite surprised as well regarding the quite scarcity of literature retrieved. However, we performed an update of the review, and this allowed us to implement the number of articles (n = 14) and reports (n = 5) included.

Let's see then:

  • The results should emphasise the most relevant information (the supplementary material shows the results in detail). Then, the main findings should be focused, providing a concise overview for the reader, perhaps by figures. Are there differences in the results obtained in scientific literature versus grey literature? Were all the bioaccessibility analyses performed on simulated biofluids? And about the grey literature (route of exposure – Supplementary table)? The different matrices influenced the bioavailability studies' results (e.g. metals, PAHs)?

Thank you for raising out this issue and allowing us to clarify this point. The summarising of the results was perhaps one of the most challenging aspect of this review. Indeed, we had to find a compromise between do not provide unnecessary details and, at the same time, highlight and value the huge variability among the included studies, preventing the complete comparison among the results. We thus decided to be a little less concise in favour of a better description of the results leading to an easier understanding of the analysis of the potential biases described in the Discussion paragraph.  However, we valued your comment and we tried to rephrase the Results paragraph. As previously described, the huge differences in experimental protocols and study design prevent the possibility to compare the results, if not in a qualitative manner. According to your suggestion of implementing a visualisation, we thus implemented the manuscript with the Figure 3 showing the proportion of studies analysing the possible release of different chemical classes.

Figure 3: Visualisation of the percentage of studies analysing the different groups of chemicals in scientific literature (outer circle) and grey literature (inner circle).

Concerning the characteristics of bioaccessibility/bioavailability, according to the inclusion criteria, we included the studies reporting data concerning the possible chemical release affecting the human organism. As reported in the results and better in the tables in Supplementary materials, the analysis was related to simulated biofliuds, except for:

  • Pronk et al. analysing also the evaporated air (as a proxy of the amount of chemicals that could be inhaled);
  • Winz et al., using a filter papers to simulate the children hand touching as a proxy of dermal contact (we thus decided to include the articles for the sake of completeness even if not specifically involving biofluids but as well a simulated support mimicking a potential route of human exposure).

We understood that the column route of exposure in the S2 table could confuse the reader, and thus we eliminated it.  

2) The discussion section should contextualise and discuss the observed results, relating them to existing knowledge. The authors showed an active role in their criticism and future research.

Thank you for the comment and for allowing us to clarify this point. We would like to highlight that we did not merely criticise the literature published thus far, but we pointed out the main weaknesses affecting the results reported, focusing on those recurrent. Aiming to overcome the complexity of the literature available, our intent was to provide useful information for those who intend to bridge the gaps still present regarding this topic. We thus not only contextualised the results, but we systematically addressed the role and the meaning of the potential biases affecting the current literature.

However, we appreciate your encouragement to give more space to the contextualization of the results and we implemented the manuscript as follows:

“To date, this is a hot topic with many research groups trying to variously highlight the possible detrimental health effects related to the exposure to these surfaces, resulting in a general call for caution [54]. However, the approaches employed in the studies implemented thus far are highly heterogeneous and there is still a stark contrast between the large body of literature assessing the presence of noxious chemicals in the components of artificial turfs and the paucity of studies systematically investigating the potential health effects associated to real-life exposure scenarios [55][47]. A recent systematic review, updated to 2023, addressed this topic, but in the last two years new articles were published re-porting some new interesting results. Moreover, this publication is only related to the scientific literature and do not include the results of the grey literature [56]. As well, the current literature analyzing the potential effects in vivo in humans are still rare and heterogeneous in terms of both chemicals and matrix under investigation. Van Rooij et al. [57] reported that the uptake of PAHs assessing the concentration of 1-hydroxypyrene in urine of seven football players training and having a match on artificial turf was minimal. Huang and colleagues, analyzing the presence of environ-mentally persistent free radicals (EPFRs)-carrying crumb rubber in saliva samples of 200 participants, revealed a potential health risk for the individuals exposed since this substance can be related to the inhibition of salivary amylases and salivary lysozyme activity [58]. Finally, Gullu et al., comparing the metals concentration (Hg, As, Al, Pb, Zn, Mg, Fe) in blood samples from 17 field hockey athletes having a tournament on synthetic turf playgrounds with 11 controls, reported that even though there some differences were observed, these were still within the international reference ranges. However, in this last case the authors did not measured the correspondent metal con-centration in the infill from the artificial playground [59]. In other cases, the biological effects measured were related to the assessment of differences in the sport performances [60]. A recent report of EPA reported the results of a biomonitoring study, measuring metals and PAHs biomarkers in blood and urine of youth and adult soccer or American football players [61]. This publication revealed blood Se levels higher than the geometric mean of the general population, even though the concentration of this substance in crumb rubber was lower than the LOD. The levels of the other metals were in line with the results for general population. Concerning the urinary PAHs, the authors observed a significant difference between pre and post-activity only for 2-hydroxynaphtalene. The creatinine adjusted levels were found to be similar to the levels observed in the general population for most of the compounds under investigation.”

3) The conclusion should briefly summarise the manuscript, emphasising its significance and suggesting future research directions (in this section, the authors recommended real exposure scenarios).

Thank you for the comment. We implemented the conclusion paragraph as follows, also in light of the new results provided by the review update:

“This review highlighted the potential release of some dangerous chemicals from crumb rubber, even though results must be considered critically, since different experimental protocols have been used and several biases could have affected the results. Moreover, the potential risk for health were in some cases reported only in worst-case scenarios, and these results are not per se sufficient to determine the risk for the sub-jects exposed. The literature analysis revealed a stark contrast between the body of literature assessing the presence of noxious chemicals crumb rubber and a small number of studies systematically investigating the potential health effects associated to real-life exposure scenarios, preventing a proper evaluation of the potential risks for human health.”

4) Line 9-10: I think that sustainability integrates environmental preservation...

Thank you for the suggestion. We implemented the manuscript as follows.

“The End-of-Life Tyres management is still a hot topic, due to implications for sustainability and human health.”

5) Line 65-66: This concern led some international Agencies involved in public health and health. Exemplify with the name of some agencies

Thank you for the comment. We updated the manuscript as requested:

“This concern led some international Agencies involved in public health and health protection (e.g., the Food and Drug Administration (FDA), and the Centers for Disease Control and Prevention (CDC), and National Institutes of Health (NIH) supporting the National Toxicology Program (NTP); the European Chemicals Agency (ECHA); the National Institute for Public Health and the Environment (RIVM); U.S. Environmental Protection Agency (EPA)) to investigate the presence of toxicants and their potentially detrimental effects [4,13,15-17].”

5) Line 102: The quality of the search strings was tested using gold standard articles. What criteria did the authors of "gold standard articles" follow? Which reliable method has been used? An exhaustive list of keywords concerning the sport surfaces and the granular infill material...

Thank you for the comment and for allowing us to clarify this point. The search strings were tested analysing their ability to retrieve specific articles (identified as “gold standard articles”) addressing the investigated topic and known before the string launch. In case of failure to find the articles retrieved, the paper would have been analysed to identify the possible reason of the failure, and the string would have been modified in order to be able to identify the gold standards. This allows the reviewers to be reasonably certain that the search strings can retrieve all the articles in literature addressing the topic under investigation.

Since this is a usual procedure to ensure the quality of the search strings, we decided to not describe it in the material and methods section to unnecessarily weigh down the manuscript.

The gold standards employed were:

Kim, S.; Yang, J.-Y.; Kim, H.-H.; Yeo, I.-Y.; Shin, D.-C.; Lim, Y.-W. Health Risk Assessment of Lead Ingestion Exposure by Particle Sizes in Crumb Rubber on Artificial Turf Considering Bioavailability. Environ Anal Health Toxicol 2012, 27, e2012005, doi:10.5620/eht.2012.27.e2012005.

Pavilonis, B.T.; Weisel, C.P.; Buckley, B.; Lioy, P.J. Bioaccessibility and Risk of Exposure to Metals and SVOCs in Artificial Turf Field Fill Materials and Fibers. Risk Analysis 2014, 34, 44-55, doi:https://doi.org/10.1111/risa.12081.

Zhang, J.; Han, I.-K.; Zhang, L.; Crain, W. Hazardous chemicals in synthetic turf materials and their bioaccessibility in digestive fluids. Journal of Exposure Science & Environmental Epidemiology 2008, 18, 600-607, doi:10.1038/jes.2008.55.

6) Line 120-124: Are the authors the independent reviewers? Who are the three independent reviewers?

Thank you for the comments. Yes, the reviewers are included in the Authors list. We updated the manuscript adding the name initials as follows:

“After the duplicate removal, two independent reviewers (F.G. and S.E.S.) performed the article screening selection in blind, at first based on titles and abstracts and then on the full-texts of the selected records. Discrepancies were discussed with a third re-viewer (G.S.).”

and

“Two independent reviewers (F.G. and S.E.S.) extracted the following data from the selected documents, and reported them in a spreadsheet: country, aim, study design, analytical methods, exposure, matrix, analysed markers, and main results.”

7) Supplementary table S.2: I propose including in the reference column not only the author's name but also the reference number.

Thank you for the comment. We updated the manuscript as requested.

10) line 154: please write: bioaccessibility measurements from scientific literature

Thank you for the comment. We updated the manuscript as requested.

“in Supplementary S.3 there is a summary of the bioaccessibility measurements from scientific literature.”

11) lines 164, 165: Among the included items, two research articles and two reports referred to data concerning playgrounds or school playgrounds [32,33,37,39], which means that the remaining articles refer to a synthetic turf rubber granule infill (e.g. KUBOTA) or turf sport fields as described in the objectives (?)

Thank you for the comment. The supposition is correct. As specified in the summarising tables in the supplementary materials, the other studies included referred to granular infill material from sport fields or destined to synthetic turf surfaces.

We updated the manuscript as follows:

“Among the included items, two research articles and two reports referred to data concerning playgrounds or school playgrounds [32,33,37,39], while the others are referred to crumb rubber from synthetic turf playground or local provider and destined to artificial turf playgrounds.”

Reviewer 3 Report

Comments and Suggestions for Authors

Dear Authors,
Thank you for the opportunity to review your manuscript on the topic "The Bioaccessibility of Chemicals in Crumb Rubber Infill Material. A Literature Review." The review systematizes existing data and raises relevant issues that are important for risk assessment and health protection. After reading the article, I have several questions and comments:

  1. Line 99 – “2. Materials and Methods” – since the presented manuscript is a Review, the manuscript structure should not include sections titled Materials and Methods. Please correct the structure of the Review according to the journal's guidelines.
  2. Line 138 – “Results” – This section title is incorrect. Please correct the structure of the Review according to the journal's guidelines.
  3. It is necessary to clarify what "grey literature" means.
  4. Figure 1 – There is no information on how the records from Google Scholar are accounted for. The diagram indicates that 89 reports were found from Google Scholar and citation searching. The authors note that these reports all proceed to text retrieval (reports sought for retrieval), all assessed (n=89), but 84 were excluded, and ultimately 9 studies (n=9) were included. However, it is unclear from where exactly these 9 were taken — from the database or from the alternative search, or both. There is no explanation for "Reports of included studies (n=5)." It is unclear what this means — whether these are reports already included in the 9 studies or additional ones, and where they came from. Within the parallelogram blocks "Identification of studies via databases and registers" and "Identification of studies via other methods," there is no final block combining data from these streams, which may be confusing.
  5. Lines 187-188 – there is a missing space between the text and the subtitle.
  6. Section “3.3 Grey literature” – for better structure, I would recommend aligning the titles and order of subsections with section “3.2. Scientific literature” and its subsections 3.2.1 to 3.2.8.
  7. Lines 307-312 – For uniformity, it is recommended to add a subsection title, as in item “3.2.1 Bioaccessibility study characteristics.”
  8. Line 480 – “6. Patents” – this section contains no text.
  9. “Supplementary S.2 Table 1 - Studies reporting bioaccessibility data in crumb rubber infill material.” – in my opinion, the table numbering should be changed since it is placed in the supplementary materials, namely to renumber it as Table S1.
  10. In my opinion, the manuscript text should include a definition of Bioaccessibility and how it is expressed in the case of Chemicals in Crumb Rubber. Probably it is even better to add a separate section dedicated to bioaccessibility in the context of studying chemicals in crumb rubber. Also, information on the bioaccessibility of chemicals contained in crumb rubber should be added to the Introduction section.
  11. In my opinion, the “Conclusions” section should not be limited to a single recommendation. This section should be expanded to reflect the study results and clearly link them to the title of the work and the stated objective.

Author Response

Comments and Suggestions for Authors - Reviewer 3

Dear Authors,

Thank you for the opportunity to review your manuscript on the topic "The Bioaccessibility of Chemicals in Crumb Rubber Infill Material. A Literature Review." The review systematizes existing data and raises relevant issues that are important for risk assessment and health protection. After reading the article, I have several questions and comments:

  1. Line 99 – “2. Materials and Methods” – since the presented manuscript is a Review, the manuscript structure should not include sections titled Materials and Methods. Please correct the structure of the Review according to the journal's guidelines.

Thank you for the comment and for giving us the opportunity to clarify this issue. As reported in the Instructions for Authors of the Journal of Xenobiotics, there is the possibility to specify the “Relevant sections”, whose title are at the discretion of the Authors. Since it is not uncommon for reviews to have a Material and Methods section we originally followed this structure. However, to comply with your request, we updated the manuscript as follows, replacing “Material and Methods” with a more general “Search strategy and literature review”

  1. Line 138 – “Results” – This section title is incorrect. Please correct the structure of the Review according to the journal's guidelines.

Thank you for the comment. The reason behind our choice in the section title was explained is the same provided in the previous answer: as well, it is not uncommon for reviews to have a Results section, and we intended this as a “Relevant Section” title.

  1. It is necessary to clarify what "grey literature" means.

Thank you for giving us the possibility to clarify this point. As reported in the Third International Conference on Grey Literature in 1997, the term grey literature consists in the information “that which is produced on all levels of government, academia, business and industry in electronic and print formats not controlled by commercial publishing”.  

To better clarify this point we added the definition into section dedicated to the search strategy.

“As reported in the Third International Conference on Grey Literature in 1997, the term grey literature consists in the information “that which is produced on all levels of government, academia, business and industry in electronic and print formats not controlled by commercial publishing”.”

  1. Figure 1 – There is no information on how the records from Google Scholar are accounted for. The diagram indicates that 89 reports were found from Google Scholar and citation searching. The authors note that these reports all proceed to text retrieval (reports sought for retrieval), all assessed (n=89), but 84 were excluded, and ultimately 9 studies (n=9) were included. However, it is unclear from where exactly these 9 were taken — from the database or from the alternative search, or both. There is no explanation for "Reports of included studies (n=5)." It is unclear what this means — whether these are reports already included in the 9 studies or additional ones, and where they came from. Within the parallelogram blocks "Identification of studies via databases and registers" and "Identification of studies via other methods," there is no final block combining data from these streams, which may be confusing.

We are sorry to find out that the Figure 1 was confusing. The synthesis is based on the flow chart form PRISMA guidelines, the section included and their order are not our choice. The selection flow process is the same for the scientific literature. The number of reports retrieved from Google/Google Scholar and citation searching were 114 (according to the number obtained from the literature review update). As specified in the text, 59 from Google/Google Scholar and 55 from citation searching. These reports were then analysed and the screening allow us to identify 5 report eligible for our review, excluding thus 109 items (as reported in the PRISMA flow chart).

  1. Lines 187-188 – there is a missing space between the text and the subtitle.

Thank you for the comment. We modified the manuscript as suggested.

  1. Section “3.3 Grey literature” – for better structure, I would recommend aligning the titles and order of subsections with section “3.2. Scientific literature” and its subsections 3.2.1 to 3.2.8.

Thank you for the comment. We modified the manuscript as suggested. Eventual differences are related to the eventual differences in the chemicals under investigation.

  1. Lines 307-312 – For uniformity, it is recommended to add a subsection title, as in item “3.2.1 Bioaccessibility study characteristics.”

Thank you for the comment. We modified the manuscript as suggested.

  1. Line 480 – “6. Patents” – this section contains no text.

Thank you for the comment. We modified according to your suggestion and we eliminated the Patents section. 

  1. “Supplementary S.2 Table 1 - Studies reporting bioaccessibility data in crumb rubber infill material.” – in my opinion, the table numbering should be changed since it is placed in the supplementary materials, namely to renumber it as Table S1.

Thank you for the comment. We modified according to your suggestion.

  1. In my opinion, the manuscript text should include a definition of Bioaccessibility and how it is expressed in the case of Chemicals in Crumb Rubber. Probably it is even better to add a separate section dedicated to bioaccessibility in the context of studying chemicals in crumb rubber. Also, information on the bioaccessibility of chemicals contained in crumb rubber should be added to the Introduction section.

Thank you for the comment. We implemented the manuscript according to your suggestion as follows:

“The term bioaccessibility can be defined as the fraction of a chemical that is potentially available for absorption, specifically gastrointestinal tract, even though this term is commonly used in literature even in a broader way [29,30].

The present review aims to describe the state of the art of the current knowledge concerning the potential release in simulated biological fluids bioaccessibility of chemicals in from the granular infill material used in artificial turf sport fields and play-grounds, considering the results in both scientific and grey literature.”

  1. In my opinion, the “Conclusions” section should not be limited to a single recommendation. This section should be expanded to reflect the study results and clearly link them to the title of the work and the stated objective.

Thank you for the comment. We modified according to your suggestion.

“This review highlighted the potential release of some dangerous chemicals from crumb rubber, even though results must be considered critically, since different experimental protocols have been used and several biases could have affected the results. Moreover, the potential risk for health were in some cases reported only in worst-case scenarios, and these results are not per se sufficient to determine the risk for the sub-jects exposed. The literature analysis revealed a stark contrast between the body of literature assessing the presence of noxious chemicals crumb rubber and a small number of studies systematically investigating the potential health effects associated to re-al-life exposure scenarios, preventing a proper evaluation of the potential risks for human health.

A holistic approach to monitoring the eventual modulation of biological outcomes as a result of the variation in the composition of artificial turf sport venues and play-grounds in real exposure scenarios would be highly recommended.”

Round 2

Reviewer 1 Report

Comments and Suggestions for Authors

Thanks for addressing the comments from the previous revision.

Author Response

Reviewer:

Thanks for addressing the comments from the previous revision.

Authors:

Many thanks for the comment. We are pleased to hear that our efforts in implementing the manuscript have fulfilled your requests and expectations.

Reviewer 2 Report

Comments and Suggestions for Authors The authors have thoroughly reviewed the entire manuscript;
the discussion and conclusion are now comprehensible.
I believe this manuscript is ready for publication.

Author Response

Reviewer:

The authors have thoroughly reviewed the entire manuscript; 
the discussion and conclusion are now comprehensible. 
I believe this manuscript is ready for publication.

Authors:

Many thanks for the comment. We are pleased to hear that our efforts in implementing the manuscript have fulfilled your requests and expectations.

Reviewer 3 Report

Comments and Suggestions for Authors

Dear authors,

Thank you for making the revisions. The manuscript is now much clearer and more complete. I have a couple of small comments left:

  1. Please add the full names for the abbreviations in lines 280-282: “27 of them were measurable in one or more than one samples. MBT, ETU and AP showed an elution rate < 10%, in contrast with BTZ, BZL, TEP, and PI showing up to 90%. Plasticisers revealed an elution rate <LOQ.”
  2. Please add the full names for the abbreviations in lines 302-304: “DMBA showed a mean bioaccessibility value of 20%, followed by DPG (16%), MBTZ (13%), DMBA (10%), CBS (5.6%), DTG (5%), IPPD (4.9%), 6PPDq (1.8%), DPPD (0.6%), and 6PPD (0.2%).”
  3. Line 306 “3.2.6 Metal(loid)s” and Line 295 “3.3.4 Metal(loids)” – Did you intentionally write them differently? If this is unintentional, please make the wording consistent.

Author Response

Reviewer:

Dear authors,

Thank you for making the revisions. The manuscript is now much clearer and more complete. I have a couple of small comments left:

Authors:

Many thanks for the comment. We are pleased to hear that our efforts in implementing the manuscript have fulfilled your requests and expectations.

Reviewer comment 1:

  1. Please add the full names for the abbreviations in lines 280-282: “27 of them were measurable in one or more than one samples. MBT, ETU and AP showed an elution rate < 10%, in contrast with BTZ, BZL, TEP, and PI showing up to 90%. Plasticisers revealed an elution rate <LOQ.”

Authors comment 1:

Thank you for raisisng out this point, allowing us to implement the manuscript. In order to not excessively complicate the manuscript text, we added the full-names of the selected compuds in the list of abbreviation at the end of the manuscript.

Reviewer comment 2:

2, Please add the full names for the abbreviations in lines 302-304: “DMBA showed a mean bioaccessibility value of 20%, followed by DPG (16%), MBTZ (13%), DMBA (10%), CBS (5.6%), DTG (5%), IPPD (4.9%), 6PPDq (1.8%), DPPD (0.6%), and 6PPD (0.2%).”

Authors comment 2:

Thank you for raisisng out this point, allowing us to implement the manuscript. In order to not excessively complicate the manuscript text, we added the full-names of the selected compuds in the list of abbreviation at the end of the manuscript.

Reviewer comment 3:

3. Line 306 “3.2.6 Metal(loid)s” and Line 295 “3.3.4 Metal(loids)” – Did you intentionally write them differently? If this is unintentional, please make the wording consistent.

Authors comment 3:

Thank you for the comment. We corrected the text as suggested.